# Nurses’ Attitudes towards Selected Social Groups: Cross-Sectional Survey among Polish Nurses

**DOI:** 10.3390/healthcare10050795

**Published:** 2022-04-25

**Authors:** Małgorzata Lesińska-Sawicka

**Affiliations:** Department of Nursing, University of Applied Sciences in Piła, 64-920 Piła, Poland; mlesinska@ans.pila.pl

**Keywords:** attitude, nursing staff, homosexual orientation, hearing-impaired person, Muslim, Roma people

## Abstract

Background: Attitude is a relatively permanent inclination towards a positive or negative evaluation of a given social or physical object, which determines a person’s disposition towards their surrounding social reality and informs his/her behavior. Aims: The aim of this study is to assess the attitudes of nursing staff, in terms of the emotional and behavioral components, in relation to selected social groups: a Roma person, a hearing-impaired person, a Muslim, and a person of a homosexual orientation. Design: This cross-sectional study was conducted by means of an Internet questionnaire. Methods: This study included 3900 nurses from Poland who were participants in social networking sites and discussion groups for nurses. The study data were collected by using a self-constructed survey questionnaire. The results were reported using the STROBE Checklist. Results: The following scale was adopted: mean 1.0–3.5—positive attitude, 3.6–6.0—negative attitude. Respondents showed positive attitudes towards patient groups (1.67–2.30), the least positive being towards Muslims (2.30) and Roma (2.21). The respondents predicted that during the performance of professional activities, they would have the biggest problem with a person of homosexual orientation (22.1%) or a Muslim person (19.0%). The results show that the age and length of service most often influence attitudes towards patients from different social groups. Conclusions: Respondents with a longer period of work experience and respondents with lower education, despite declaring positive attitudes towards the surveyed social groups, expressed negative statements towards Muslims and homosexuals. Cultural education during the undergraduate and postgraduate studies of nursing staff is essential. Impact Statement: This research indicates that the lesser the need for direct involvement in interactions with patients from other groups, the greater the willingness to accept the situation in which care is provided.

## 1. Introduction

Different definitions of the concept of attitude may be encountered in the literature. They stem from sociological [1,2,3] and psychological [4,5,6] concepts and are based on the behaviorist tradition or the psychology of learning [7,8]. The common denominators for defining the concept of attitude are as follows: a specific object of reference, which can be physical or social in nature, the tendency to behave in a certain way, and the associated emotions and cognitive processes.

Generally speaking, attitude determines a person’s disposition towards their surrounding reality and informs his/her behavior.

Attitude is shaped by external factors, such as socio-cultural conditioning, and internal factors, such as personal experiences. It can also be influenced by information from other people, observations, or the mass media [9].

One of the properties of attitude is its direction, also called the attitude sign, which makes it possible to distinguish between positive and negative attitudes towards a given object. It concerns the emotional–appreciative component, i.e., the feelings that someone has towards the attitude object. If they are positive, they feel a pull towards this object; if negative, they tend to avoid it [10,11].

In their professional work, nurses interact with patients from different social groups, which can be distinguished by many categories, e.g., gender, nationality, religion, or type of physical dysfunction. Each group is characterized, among other things, by its own peculiarities: it confers a social identity, creates a “we” consciousness and a sense of separateness, and defines a concept of values common to its members.

This cultural difference can be a source of aversion, fear and insecurity or curiosity, admiration, and fascination. Seeing the “other” as interesting, stimulating and valuable fosters interactions based on mutual recognition and respect and directs interactions towards cooperation and collaboration. When the “other” is perceived as insignificant, it triggers aggressive behavior, antagonism, and domination [12].

Inherent in nursing is a respect for human rights, including cultural rights, the right to life and choice, the right to dignity and the right to be treated with respect. According to the Code of Ethics of the International Council of Nurses, nursing care should not be restricted because of age, color of skin, religion, culture, disability or illness, gender, sexual orientation, nationality, political opinion, race, or social situation. Nursing staff are trained to provide health care to all individuals, families, and communities [13].

Specific attitudes of nurses towards a patient from a different social group, or a different cultural background, may provide the patient with a feeling that his/her rights are respected, and he/she is treated with kindness, fairly and in accordance with his/her expectations, or quite the reverse—without respect or understanding, with distance, or based on stereotypes. These attitudes will determine actions that may facilitate cooperation with the patient and his/her faster recovery and independence, or will cause disappointment, discomfort, incomprehension, and loneliness.

In the literature, one can find the described attitudes of nursing students or social workers towards different social groups, such as the elderly, pregnant teenagers, or people with disabilities [14,15,16,17,18,19]. To date, the author of this manuscript has not found in the literature studies that show a comparison of the attitudes of nurses towards different social groups, which would make it possible to examine what attitudes, being more or less positive, occur in nurses, thus influencing the nature of the interaction between them and their patients. Due to the lack of such studies, there is a certain gap which, if unnoticed and not filled with reliable knowledge, may reduce the level of preparation of graduates for professional work and their ability to cope with difficult situations, as well as affect the quality of the professional work of nurses. It can also result in patients being unequally treated, leaving them with anxious needs [20].

The aim of this study was to assess the attitudes of nursing staff towards selected social groups, different in terms of sexual orientation, nationality, religion, and physical functioning.

The groups selected for this study are social minorities in Poland characterized by specific features that distinguish them from the majority of society. A minority is a group of people in a country who differ from the majority of its citizens by virtue of their nationality, race, religious affiliation, spoken language, traditions, customs, or views on certain issues, etc. [21].

Approximately 41,000 people in Poland are followers of Islam [22]. During the National Census of Population and Housing conducted in 2011, a Roma affiliation was declared by 16,725 Polish citizens [23]. According to statistics from the Polish Association of the Deaf, there are approximately 800,000 people with hearing impairments in Poland, including approximately 50,000 deaf people who use Polish sign language to communicate and have difficulty understanding text in Polish [24].

Each of the groups listed is accompanied by a different context that may involve the attitudes of nursing staff toward its representatives.

According to the World Health Organization, hearing loss is now the fourth greatest contributor to years lost to disability globally [25]. People who are deaf or hard of hearing use the sign language system to communicate with others. This different method of communication makes them a specific group of patients for whom additional care skills are required. This makes it difficult for nursing staff who do not know sign language to communicate with such patients [26].

A review of the literature shows the negative attitudes of medical workers [27] or nursing students towards homosexuals [28], despite the fact that the need to know and respect the rights of sexual minorities is being increasingly emphasized [29]. The attitudes of medical professionals towards homosexuals can influence their willingness to provide these individuals with medical help [30].

The Roma are an ethnic group who have lived in Europe since their migration from India over 1000 years ago. Roma cultural heritage includes a rich oral tradition, art forms such as flamenco, an emphasis on family, and *Romanës*, the Roma language. The Roma are among the most disenfranchised, socially unaccepted, and morally vilified ethnic minority groups in Europe and especially in East-Central European countries [31,32]. As a culturally and linguistically diverse group, Roma people are portrayed as beggars, criminals, profiteers, and lazy, being a target of marginalization and social exclusion, as well as perpetual discriminatory and violent practices on an interpersonal, institutional, and national level [33].

Islam is becoming an increasingly prevalent religion in Europe due to large inflows of Muslims over the last few decades [34]. With the growing Muslim population, Islam is becoming an increasingly important religion in Europe, and in many Western European countries, Islam constitutes the second largest religion after Christianity [35]. The European public is critical of immigration from Muslim countries [36]. The increasing presence of Muslims in Europe has led to a variety of debates which portray Muslims as a threat to the West, and the terrorist attacks in Europe have intensified these debates and have led to increasing Islamophobic discourse and incidents in Europe [37]. According to Europol’s 2021 EU Terrorism Situation Report, there were 57 attempted terrorist attacks in the EU in 2020 (including successful, failed, and foiled attacks), compared with 55 in 2019. Ten of these were attributed to jihadist terrorism in Austria, France, and Germany. Although jihadist terrorists were only behind one-sixth of all attacks in the EU, they were responsible for more than half of the deaths and almost all of the injuries. The total number of deaths and injuries in the EU doubled from 10 deaths and 27 injuries in 2019 to 21 deaths and 54 injuries in 2020 [38]. As a result, Islam and Muslims have often been portrayed through the discourse of violence, with parallels drawn between terrorism and immigration [34].

Showing the attitudes of nursing staff towards selected social groups, differing in sexual orientation, nationality, religion, and physical functioning, will help to understand their different attitudes towards minority patients.

## 2. Materials and Methods

### 2.1. Study Design

A cross-sectional study design was used.

### 2.2. Participants

The study involved 3921 nurses working in Polish health care institutions who agreed to participate in the study. After analyzing the collected data for data complementarity, 3900 respondents were finally qualified for the study. No incentive was used to encourage participation in the study. Most of the respondents were aged 20–30 years (40.0%), had an undergraduate, bachelor’s degree (53.3%), and had worked in the profession for 1–5 years (37.9%) (Table 1).

### 2.3. Procedure/Course of the Study

The main study was preceded by a pilot study outside the area of the present research. Links to the survey were sent out electronically. After the pre-testing of the questionnaire was completed, the researcher modified some questions (for better clarity and wording) based on feedback from the pre-test. Cronbach’s alpha coefficient (α = 0.85) was calculated, providing feedback on the reliability of the scale.

Data collection took place between April and July 2021. The study was conducted using a survey distributed as a link posted on social networking sites and discussion groups for nurses from Poland. The link was posted with the permission of group administrators and moderators. Before completing the survey, respondents were required to declare their consent to participate in the study. The participants, together with the link to the questions, received information about the procedure and purpose of the survey, and that by returning the answers to the questions, they were also consenting to data processing.

The time taken to respond and return the answers was 15 min.

To minimize the possibility of the same person participating in the survey on multiple occasions, the survey started with a welcome screen with the title, the purpose of the survey, instructions, and a link to the survey. In this way, the potential respondent knew immediately what the topic of the survey was and could leave before starting to answer the questions.

Web-based electronic survey software (Google Forms) was used to collect the data. After data collection, each questionnaire was visually checked for completeness.

### 2.4. Measures/Tool

A questionnaire form prepared by the researchers was used as a data collection tool. The choice of standardized interviewing as a research technique was related to the desire to collect homogeneous and comparable data.

For the purposes of this study, the estimation method was used, using questions from the Bogardus Scale [39], concerning the respondents’ willingness to enter into different interpersonal/social situations with the object:

1—I would not exclude a Roma/a person with a different sexual orientation/a hearing-impaired person/a Muslim from my country;

2—I would accept a Roma/a person with a different sexual orientation/a hearing-impaired person/a Muslim as guest in my country;

3—I would accept a Roma/a person with a different sexual orientation/a hearing-impaired person/a Muslim as a resident in my country;

4—I would accept a Roma/a hearing-impaired person/a Muslim/a person with a different sexual orientation as a co-worker;

5—I would accept a Roma/a hearing-impaired person/a Muslim/a person with a different sexual orientation as a neighbor on my street;

6—I would accept a Roma/a hearing-impaired person/a Muslim/a person with a different sexual orientation as a close friend;

7—I would accept a Roma/a hearing-impaired person/a Muslim/a person with a different sexual orientation as a spouse of a close relative or friend.

When choosing the best descriptor for a given question, respondents could choose the following answers: definitely yes, yes, rather yes, rather no, no, and definitely no, with scores from 1 (strongly agree) to 6 (strongly disagree).

In addition to sociometric data, the study contained questions allowing the respondents to self-assess their personal tolerance, if they are swayed by stereotypes in their professional life, the emotions they feel while caring for patients from other social groups and the possibility of establishing professional relationships with them.

### 2.5. Statistical Analysis

Data were entered, cleaned, and coded using Statistica 8.0 PL software. Data were screened for outliers and the assumptions of parametric tests. Based on the answers obtained, the average of all questions was calculated. A scale was adopted: average 1.0–3.5—positive attitude; 3.6–6.0—negative attitude.

The results are presented as descriptive statistics and a frequency table. The frequency distribution for each variable was determined, and the mean (M) and standard deviation (SD) were calculated, with an assumed significance level of *p* = 0.05. Student’s t-test was used to assess the statistical significance of differences in the mean scores of attitudes in the declarations of respondents differentiated by gender, education, and length of service in relation to specific social groups. The single-sample chi-square test of variance, ANOVA, and a post hoc test (Fisher LSD) were used to evaluate the existence of differences between the groups.

### 2.6. Ethical Considerations

Before conducting the research, the necessary consent of the Ethics Committee of the University of Applied Sciences in Piła was obtained. Respondents were informed that they could withdraw from the study without giving any reasons and without any consequences; that their responses would be anonymized by removing any personal information and the computer ID and would be analyzed with other responses to obtain aggregate results; no identifying information would be included in this dataset; and there was no direct personal benefit associated with participation in this study.

## 3. Results

The respondents mostly considered themselves as tolerant (98.4%), and not swayed by stereotypes in their professional life (83.1%) (Table 2).

Analyzing the mean of the respondents’ statements regarding their agreement to enter into different interpersonal and social situations with a selected social group, it can be stated that all of them fall within the range of attitudes assumed at the beginning of the research to be positive attitudes, with the mean ranging from 1.67 towards people with hearing impairment to 2.30 towards Muslims. Analyzing the SD score, it can be concluded that more extreme opinions were given by the respondents towards Muslims (Figure 1).

On the basis of the results obtained, it can be stated that with less need for direct involvement in an interaction with representatives of a culturally different group, the greater the willingness to accept a given situation. The overall mean for question 1—I would not exclude a Roma/a person with a different sexual orientation/a hearing-impaired person/a Muslim from my country—was 1.95, while for question 7—I would accept a Roma/a person with a different sexual orientation/a hearing-impaired person/a Muslim as a spouse of a close relative—it was 2.25. Both results are within the accepted boundaries recognized as positive attitudes but possess differences which are statistically significant (*p* = 0.01) (Table 3).

An in-depth analysis of the respondents’ statements showed a statistically significant relationship between the social group, variables such as age, seniority, and level of education, and the type of question (Table 4). Among the sociometric data, only the level of education did not show any influence on attitudes towards persons of a different culture. Statistical analysis showed the existence of the largest number of correlations between the views presented by the respondents and persons of a homosexual orientation (questions 1, 4 and 7) in relation to age and seniority. The existence of statistical dependencies with regard to persons of a Roma concerned question 1, of persons with hearing impairment—questions 1 and 2, and of Muslims—questions 1 and 6.

The results in Table 5 show that the respondents would not accept a homosexual as a close relative, and in the case of Muslim persons, the lack of acceptance concerned all the situations discussed in the study.

ANOVA analysis for independent groups and a post hoc test (Fisher LSD) showed differences between groups (Table 6) (Figure 2). In questions 1, 4, 6 and 7, Roma people and Muslims were more likely to receive negative opinions than deaf people and homosexuals.

In question 2, homosexuals and Roma people were more likely to receive negative opinions than deaf people and Muslims. In question 5, homosexuals and deaf people were more likely to receive positive opinions than Muslims and Roma people.

In the opinion of the respondents, the cultural diversity of patients does not matter to them when it comes to difficulty in establishing professional relationships with the representatives of selected social groups. According to the respondents, it is the easiest to establish a professional relationship with Roma persons (18.5%) and the most difficult with persons of homosexual orientation (22.1%) (Table 7).

Positive emotions such as empathy, sympathy, and understanding were mentioned most frequently among the emotions that might accompany respondents during their professional work towards representatives of particular social groups. Just a few individuals indicated negative emotions towards Muslims, Roma persons, and persons of homosexual orientation (Table 8).

## 4. Discussion

Many contemporary European societies, not only Poland, are no longer homogeneous societies in terms of dimension and understanding, whose internal cohesion is created by a uniform identity, history, axiology, and culture. The ease of movement between countries, different geographic areas, the increasing liberality towards people from the LGBTQ+ community, the increasingly better technology that facilitates the functioning of people with different physical dysfunctions, etc., mean that streets, hospitals, schools, and other places of public use are filled with people from different social groups, people who are diverse in every way.

Today, health care professionals around the world provide care for an increasing number of culturally diverse patients [40].

The attitude of nursing staff towards their patients is one of the components of professional competence, including cultural competence. Attitude guides nurses’ interactions, and educational, diagnostic, and preventive treatments. The results of this study indicate that nursing staff hold positive attitudes towards different groups of patients. This is a good indicator and forms a solid basis for communicating and interacting with patients, including people from culturally different backgrounds or social groups.

Positive contact between members of different groups is considered one of the most effective ways to reduce prejudice [41]. However, as highlighted by intergroup contact researchers [42,43], intergroup encounters in everyday life can be perceived as unpleasant, unfriendly, and anxiety-producing (negative intergroup contact). In fact, positive and negative inter-group contact are discrete experiences, rather than two opposite poles of a continuum [44,45]. An individual may have several encounters with members of another group, and some of these encounters are experienced as positive and pleasant, while others are perceived as negative. Positive and negative contact experiences are independent predictors of intergroup prejudice and attitudes. While positive contact reduces prejudice, negative contact increases it [46].

The group towards which respondents showed the most positive attitude was hearing-impaired persons. Although respondents showed empathy, compassion, or understanding towards this group of people, they also felt uncomfortable and indicated that they experienced the greatest difficulty in establishing professional relationships with this group of patients, compared with all groups mentioned in this study. In addition, hearing-impaired patients are sometimes judged to be of lower intelligence because of their difficulty in understanding messages [47]. This can cause the belittling of these patients, but may also evoke positive emotions and attitudes, albeit with negative undertones, such as pity. An effective knowledge of sign language can facilitate proper communication with hearing-impaired patients, as well as assessment and action oriented to the patients’ expectations [48,49], thus building a correct image of the patient and adopting an adequate nursing attitude.

Another group towards which the respondents showed positive attitudes was those of homosexual orientation.

Persons of homosexual orientation evoke a range of different emotions. The respondents taking part in the study showed sympathy and empathy, but some also felt embarrassment, irritation, anger/rage, hatred or contempt, and repulsion. Emotional aspects in some people appear as non-specific dread or discomfort associated with homosexuality and homosexual people. Some may react with guilt, shame, awkwardness, embarrassment, and fright, while others experience more hostile feelings such as anger, disgust, and disdain [19]. Combating homophobia or unfavorable attitudes towards patients with a different sexual orientation is an important issue addressed in both the pre- and postgraduate education of medical personnel [50]. According to the Eurobarometer, the support in the EU for LGBTIQ equality increased from on average 71% of EU citizens in 2015 to 76% in 2019. Despite this increase of 5% of EU citizens who support LGBTIQ equality, there is still a wide divergence at the Member State level where support ranges from 31% of citizens in Slovakia to 98% in Sweden [51].

The Roma are a minority around which many stereotypes and prejudices have accumulated. Researchers have argued that anti-Roma attitudes are a unique form of prejudice [52]. Roma people are usually portrayed as beggars, criminals, profiteers, and lazy, a target of marginalization and both social and perpetual exclusion [33,53,54]. The results of a representative study conducted in Hungary, Romania, Slovakia, France, and Ireland by Nariman and others showed the presence of empathy towards the Roma in Hungary, the perception of a threat to national identity in Romania, and sympathy in Slovakia, France, and Ireland. They showed that stronger negative attitudes towards Roma, as well as stereotypical evaluations, are significantly linked to the historical background of a country [33].

Despite negative attitudes found in the literature, respondents to this survey showed positive attitudes towards the Roma. Roma culture is often misunderstood by other social groups, hence the prejudices and negative attitudes that arise. Learning about the basics of Roma behavior would certainly improve the attitudes and perceptions of nursing staff towards this group.

The highest percentage of extreme statements, although still within the range of positive attitudes adopted in this survey, was presented by the respondents towards Muslims. Hostile attitudes towards Muslims can be observed in practically every European country [55]. There are various explanations as to why anti-Muslim attitudes arise. One such example is the way Muslims and Islam are portrayed by the media. Muslims and Islam are often portrayed in a negative light, and “Islamic culture” is sometimes presented as the complete opposite of “Western culture” [56]. This is often accomplished by showing images of bombings, destroyed buildings, and killings in the name of Islam [57]. Exposure to news related to Muslims has been found to influence how Muslims are perceived [58]. Guscito and others have observed polarized perceptions of Muslims in Europe. Central and Eastern European countries are much more opposed to further Muslim immigration compared with Northern and Western European countries. This is due to the fact that Eastern European natives have a stronger sense of self-identification than Western European natives, who do not perceive Muslims as a threat to the same extent as respondents in Eastern/Central Europe due to the significantly larger number of Muslims in Western European countries [34]. Misunderstandings of cultural differences and the use of stereotypes when caring for Muslims are addressed in the literature [58], while also indicating the need for nursing staff to increase their knowledge in order to improve service delivery while respecting cultural distinctiveness.

Negative attitudes, resentment, and prejudice of health professionals towards patients from culturally different groups can lead to limitations in the provision of quality medical care, as well as higher rates of morbidity and mortality among members of these groups.

The findings show that with less direct involvement in an interaction with a culturally different group, the greater the tendency to accept the situation. This is a somewhat false understanding of acceptance and tolerance towards culturally different people. As long as the situations are ‘distant’ and do not involve face-to-face contact, people are inclined to make judgements about their attitudes as positive. The closer the contexts of interpersonal relationships and social situations become, the more frequent is the distancing.

### Limitations

The conducted research, while providing important knowledge, has several limitations. Thanks to the Internet, reaching respondents from all walks of life is easier, simpler, faster, and less costly, as the questionnaires do not have to be printed or distributed directly. However, conducting research with the use of the Internet is associated with various difficulties, including technical and practical as well as substantive and methodological issues. First, the availability of online survey is limited. Therefore, the results cannot be generalized to nurses in Poland who do not use social networking sites and discussion groups. Another limitation of the study presented here is the relatively small number of respondents over 50 years of age, which may somewhat distort the true picture of the entire nursing population, as this age group is the most numerous among nurses in Poland. The average age of nurses in Poland in 2020 was 53 years. The older the person, the less often he or she uses the Internet. The possibility of the repeated participation of the same person in the study is also a difficulty.

Due to the personal nature of the survey questions, there is the possibility of information error related to the survey format. This may lead to inaccurate responses, and this analysis depends on the credibility of the responses. However, the author hopes that the anonymous and online format of this questionnaire will help to obtain a greater accuracy of answers than direct contact with the researcher.

In addition, the research presented here deals with respondents’ general attitudes towards culturally different patients; the cognitive component was not included. Most of the respondents’ assessments concern their declarations and perceptions of such interactions. In reality, behavior may differ from the opinions presented by the respondents. Respondents’ opinions are based either on individual professional or personal encounters or on generalized, colloquial knowledge.

## 5. Conclusions

Respondents declared positive attitudes towards different groups of patients. The most frequent negative answers appeared in relation to Muslims and homosexuals, although they fell within the range of positive attitudes. Respondents anticipated that they may have the biggest problem with these groups during their professional activities. The research indicates that with less necessity to be directly involved in interactions with patients coming from other groups, the higher the willingness to accept the situation.

The respondents declared that the social origin of patients does not matter to them when establishing professional relationships. However, the research results indicate that age and length of service influence attitudes towards culturally different people. This should be an indication and rationale for starting cultural education from an early age and continuing it during the pre- and postgraduate training of nursing staff. There is a need for ongoing training in cultural competence.

## Figures and Tables

**Figure 1 healthcare-10-00795-f001:**
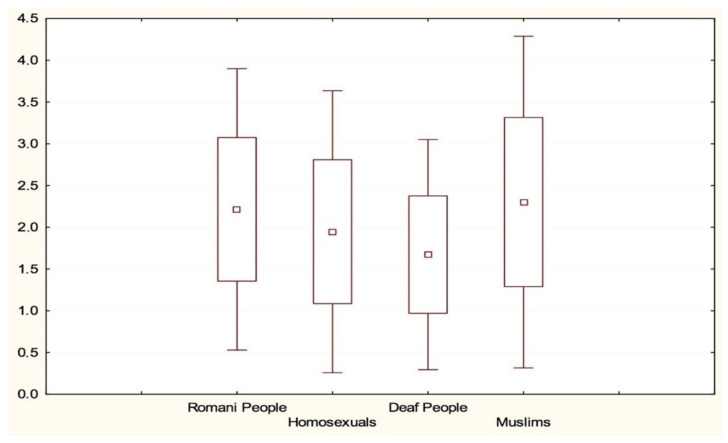
Mean indications of respondents towards social groups.

**Figure 2 healthcare-10-00795-f002:**
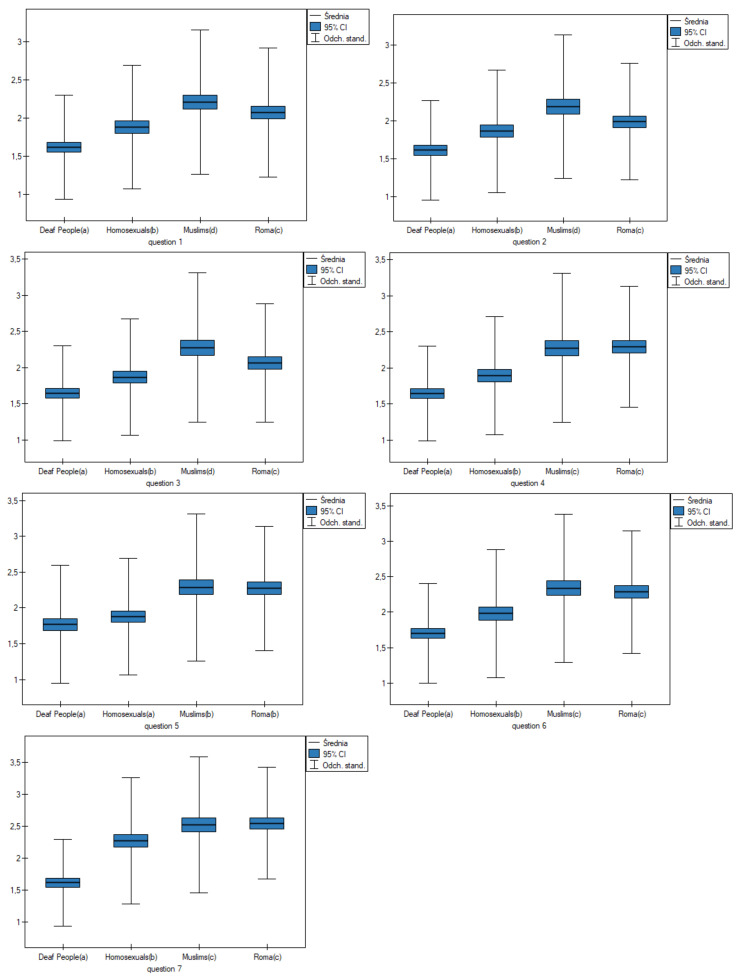
Observed variation in results between groups.

**Table 1 healthcare-10-00795-t001:** Sociodemographic characteristics of the study population.

Sociometric Variable	N (%)
Age	
20–30 years	1560 (40.0%)
31–40 years	940 (24.1%)
41–50 years	1040 (26.7%)
51 years and older	360 (9.2%)
Education	
Secondary	240 (6.2%)
Post-secondary	100 (2.6%)
Bachelor degree	2080 (53.3%)
Master’s degree	1480 (37.9%)
Seniority	
1–5 years	1480 (37.9%)
6–10 years	640 (16.4%)
11–15 years	460 (11.8%)
16 years and more	1320 (33.9%)

**Table 2 healthcare-10-00795-t002:** Self-assessment of respondents as tolerant and not swayed by stereotypes in their professional life.

Sociometric Variable	N (%)
Self-assessment as a tolerant person	
Definitely yes	1280 (32.8%)
Yes	1590 (41.0%)
Rather yes	950 (24.1%)
Rather not’	20 (0.5%)
No	60 (1.6%)
Definitely not	-
Swayed by stereotypes in professional life	
Definitely yes	200 (5.1%)
Yes	200 (5.1%)
Rather yes	260 (6.7%)
Rather not	1660 (42.6%)
No	1220 (31.3%)
Definitely not	360 (9.2%)

**Table 3 healthcare-10-00795-t003:** The mean of the respondents’ answers to particular questions in relation to social group.

Social Group Category	Question’s Number							Statistic	
	1	2	3	4	5	6	7	M	SD
	Average of Answers								
Ethnic group/ nationality	2.06	1.98	2.06	2.27	2.28	2.28	2.50	2.21	0.86
Sexual orientation	1.87	1.86	1.86	1.87	1.89	1.97	2.27	1.94	0.86
Physical fitness	1.61	1.61	1.62	1.77	1.65	1.70	1.72	1.67	0.70
Religion	2.20	2.18	2.29	2.28	2.27	2.33	2.50	2.30	1.01
Total	1.95	1.91	1.89	2.04	2.02	2.07	2.25	-	

**Table 4 healthcare-10-00795-t004:** Analysis of social group, type of question and sociometric data.

Category	Variables	Statistic Test	Question’s Number						
			1	2	3	4	5	6	7
Ethnic group/ nationality	Age	t-value	−1.56	0.49	−1.63	−1.13	−0.23	−1.15	−0.40
		*p*	0.12	0.62	0.11	0.87	0.87	0.78	0.68
	Education	t-value	0.13	1.03	1.14	1.23	1.50	1.13	0.87
		*p*	0.90	0.30	0.26	0.22	0.89	0.26	0.39
	Seniority	t-value	−2.37	−1.44	−1.23	−0.59	0.93	−0.40	−1.13
		*p*	0.02	0.15	0.22	0.56	0.35	0.69	0.26
Sexual orientation	Age	t-value	−2.39	−1.05	−1.32	−1.98	−1.89	−1.83	−2.31
		*p*	0.02	0.30	0.19	0.05	0.06	0.07	0.02
	Education	t-value	0.42	1.27	1.54	1.54	1.54	1.57	1.27
		*p*	0.67	0.21	0.13	0.13	0.13	0.12	0.21
	Seniority	t-value	−2.99	−1,71	−1.89	−1.99	−1.96	−1.79	−2.18
		*p*	0.00	0.09	0.06	0.04	0.05	0.08	0.03
Physical fitness	Age	t-value	−2.54	−2.03	−1.39	−1.35	0.81	−1.78	1.78
		*p*	0.01	0.04	0.17	0.18	0.41	0.18	0.23
	Education	t-value	1.80	1.17	0.97	1.77	1.65	1.04	1.63
		*p*	0.28	0.24	0.33	0.29	0.30	0.30	0.29
	Seniority	t-value	−1.40	−0.56	−0.49	−0.40	−0.60	−0.51	−0.67
		*p*	0.17	0.58	0.63	0.69	0.55	0.61	0.51
Religion	Age	t-value	−1.94	−1.29	−1.54	−1.68	−1.89	2.67	−1.56
		*p*	0.06	0.20	0.14	0.10	0.06	0.04	0.12
	Education	t-value	−0.02	1.05	1.02	0.96	1.20	−0.73	1.21
		*p*	0.10	0.30	0.31	0.34	0.24	0.47	0.22
	Seniority	t-value	−2.18	−1.45	−1.27	−1.30	−1.37	−1.40	−1.56
		*p*	0.03	0.15	0.21	0.20	0.17	0.17	0.12

**Table 5 healthcare-10-00795-t005:** Relationships between the respondents’ acceptance in the questions and the social group.

Variables	Question’s Number						
	1	2	3	4	5	6	7
Roma	*p* < 0.001	*p* < 0.001	*p* < 0.001	*p* < 0.001	*p* < 0.001	*p* < 0.001	*p* < 0.001
Person with homosexual orientation	*p* < 0.001	*p* < 0.001	*p* < 0.001	*p* < 0.001	*p* < 0.001	*p* = 0.005	*p* = 0.686
Deaf Person	*p* < 0.001	*p* < 0.001	*p* < 0.001	*p* < 0.001	*p* < 0.001	*p* < 0.001	*p* < 0.001
Muslim	*p* = 0.129	*p* = 0.132	*p* = 0.895	*p* = 0.297	*p* = 0.491	*p* = 0.223	*p* = 0.069

**Table 6 healthcare-10-00795-t006:** Summary of interaction effects between groups in each question.

Question		Deaf Person	Homosexual	Muslim	Roma	F
			*p*			
1	Deaf Person		0.001	<0.001	<0.001	*p* < 0.001
	Homosexuals	0.001		<0.001	0.001	
	Muslims	<0.001	<0.001		0.022	
	Roma	<0.001	0.001	0.022		
2	Deaf Person		0.001	<0.001	<0.001	*p* < 0.001
	Homosexual	0.001		<0.001	0.028	
	Muslim	<0.001	<0.001		0.002	
	Roma	<0.001	0.028	0.002		
3	Deaf Person		0.001	<0.001	<0.001	*p* < 0.001
	Homosexual	0.001		<0.001	0.001	
	Muslim	<0.001	<0.001		0.001	
	Roma	<0.001	0.001	0.001		
4	Deaf Person		0.001	<0.001	<0.001	*p* < 0.001
	Homosexual			<0.001	<0.001	
	Muslim	<0.001	<0.001		0.823	
	Roma	<0.001	<0.001	0.823		
5	Deaf Person		0.089	<0.001	<0.001	*p* < 0.001
	Homosexual	0.089		<0.001	<0.001	
	Muslim	<0.001	<0.001		0.839	
	Roma	<0.001	<0.001	0.839		*p* < 0.001
6	Deaf Person		0.001	<0.001	<0.001	
	Homosexual	0.001		<0.001	0.001	
	Muslim	<0.001	<0.001		0.403	
	Roma	<0.001	0.001	0.403		
7	Deaf Person		<0.001	<0.001	<0.001	*p* < 0.001
	Homosexual	<0.001		0.001	0.001	
	Muslim	<0.001	0.001		0.708	
	Roma	<0.001	0.001	0.708		

**Table 7 healthcare-10-00795-t007:** Difficulty in establishing professional relations with representatives of selected social groups in the respondents’ opinion.

Variables	The Easiest to Establish a Working Relationship N (%)	The Most Difficult to Establish a Working Relationship N (%)
Roma	780 (18.5%)	140 (3.6%)
Person with homosexual orientation	420 (10.8%)	860 (22.1%)
Deaf Person	120 (3.1%)	220 (5.6%)
Muslim	20 (2.0%)	740 (19.0%)
Does not matter	2560 (65.6%)	1940 (49.7%)

**Table 8 healthcare-10-00795-t008:** Anticipated types of emotions indicated by the respondents while performing their professional duties towards particular social groups.

Emotions	Roma N (%)	Person with Homosexual Orientation N (%)	Deaf Person N (%)	Muslim N (%)
Sympathy	650 (16.6%)	1060 (27.1%)	870 (22.3%)	680 (17.4%)
Empathy	910 (23.3%)	1030 (26.4%)	1120 (28.7%)	960 (24.6%)
Compassion	270 (6.9%)	300 (7.7%)	610 (15.6%)	230 (5.9%)
Indifference	470 (12.1%)	250 (6.4%)	250 (6.4%)	540 (13.8%)
Understanding	960 (24.6%)	1000 (25.6%)	860 (22.1%)	920 (23.6%)
Embarrassment	210 (5.4%)	60 (1.6%)	140 (3.6%)	180 (4.6%)
Fear	210 (5.4%)	60 (1.6%)	50 (1.3%)	240 (6.2%)
Irritation	120 (3.1%)	60 (1.6%)	0	60 (1.5%)
Anger/Rage	40 (1.0%)	20 (0.5%)	0	30 (0.8%)
Hatred	20 (0.5%)	20 (0.5%)	0	10 (0.3%)
Contempt	20 (0.5%)	20 (0.5%)	0	30 (0.8%)
Repulsion	20 (0.5%)	20 (0.5%)	0	20 (0.5%)

## Data Availability

The data analyzed in the study are available upon request to the first author.

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
