# Peer review of "Nurses’ Attitudes towards Selected Social Groups: Cross-Sectional Survey among Polish Nurses"

_healthcare, 2022, doi:10.3390/healthcare10050795_

Round 1
Reviewer 1 Report
The article: Nurses’ Attitudes Towards Selected Social Groups. Cross-sec-2 tional survey among Polish nurses, contains important elements for assessing the attitudes of nurses. The aim of this study was to assess the attitudes of nursing staff, the emotional and behavioural component, in relation to selected social groups: a Roma people, a hearing impaired person, a Muslim, a person with a homosexual orientation. It is a pity that only these four groups, because it was possible to study other groups, e.g. the homeless or the Jews.
Comments:
- the abstract contains the repetition of 3 sentences: "The study 15 included 3900 nurses from Poland who were participants in social networking sites and discussion 16 groups for nurses. The study data were collected by using the a self-constructed survey question-17 naire. Results were reported using the STROBE Checklist.";
- the type of variables and the statistical analysis are quite poor;
- little results from other countries in the discussion;
- well defined study limitations.
Author Response
Dear Reviewer,
Thank you for your suggestions. I have tried to adapt to them.
I hope the manuscript looks better now. I have highlighted in yellow all the changes made.
The article: Nurses’ Attitudes Towards Selected Social Groups. Cross-sec-2 tional survey among Polish nurses, contains important elements for assessing the attitudes of nurses. The aim of this study was to assess the attitudes of nursing staff, the emotional and behavioural component, in relation to selected social groups: a Roma people, a hearing impaired person, a Muslim, a person with a homosexual orientation.
Thank you for your opinion and comments. I have tried to make the best of your suggestions. I hope the manuscript looks better now.
It is a pity that only these four groups, because it was possible to study other groups, e.g. the homeless or the Jews.
Thank you for the suggestion. The intention of the study was to show the attitudes of nurses towards selected four groups: a group with different sexual orientation, a group with different religion, a group with different country of origin and a group with different physical functioning. In the future, the study can be extended to other aspects, e.g. attitudes towards patients from different religions, etc. International or inter-centre studies could also be conducted.
Comments:
- the abstract contains the repetition of 3 sentences: "The study 15 included 3900 nurses from Poland who were participants in social networking sites and discussion 16 groups for nurses. The study data were collected by using the a self-constructed survey question-17 naire. Results were reported using the STROBE Checklist.";
Thank you for drawing attention to this oversight. The repeated sentences have been removed.
- the type of variables and the statistical analysis are quite poor;
Statistical analyses have been added, such as Chi-square test of single sample variance, ANOVA and post hoc (Fisher LSD)
- little results from other countries in the discussion;
The discussion has been supplemented by the results of studies from other countries. Thank you for your suggestions. This has given the discussion more depth.
- well defined study limitations
Thank you
Reviewer 2 Report
I have read the article entitled "Nurses’ Attitudes Towards Selected Social Groups. Cross-sectional survey among Polish nurses". The article is an interesting study. However, it requires elaboration of the content. Please check the content of the abstract carefully due to the repetition of the content. Also, the content indicated in the limitations of the study in lines 286-292 in my opinion has nothing to do with the limitations of the study and should be included in the introduction. In the limitations of the study, it is also worth referring to the tool and method of data collection. The dissemination of the link to the research on various forums and profiles at the same time could cause the questionnaire to be completed several times by the same person. How did the author solve this issue? It is worth pointing to this.
Author Response
Dear Reviewer,
Thank you for your suggestions. I have tried to adapt to them.
I hope the manuscript looks better now. I have highlighted in yellow all the changes made.
I have read the article entitled "Nurses’ Attitudes Towards Selected Social Groups. Cross-sectional survey among Polish nurses". The article is an interesting study. However, it requires elaboration of the content.
Thank you for your opinion and comments. I have tried to make the best of your suggestions. I hope the manuscript looks better now.
Please check the content of the abstract carefully due to the repetition of the content.
Thank you for drawing attention to this oversight. The repeated sentences have been removed.
Also, the content indicated in the limitations of the study in lines 286-292 in my opinion has nothing to do with the limitations of the study and should be included in the introduction.
Thank you for your suggestion. I corrected this part of manuscript
In the limitations of the study, it is also worth referring to the tool and method of data collection.
Thank you very much for this valuable comment. I have added in the limitations the specificity of research conducted via the Internet.
The dissemination of the link to the research on various forums and profiles at the same time could cause the questionnaire to be completed several times by the same person. How did the author solve this issue? It is worth pointing to this.
I added: To avoid multiple participation in the survey posted on social networking and discussion groups, the title of the survey was provided at the beginning of the questionnaire. The title was displayed together with a description and a link to the questions. This way the potential respondent knew immediately what the topic of the survey was.
Reviewer 3 Report
This study aims to assess Nurses’ Attitudes Towards Selected Social Groups by using a cross-sectional survey among Polish nurses.
- In the paper title, consider a colon between the two parts of the title. For instance: ‘Nurses’ Attitudes Towards Selected Social Groups: Cross-sectional survey among Polish nurses’.
- Line 15 – Check spelling in “sur-vey”
- Lines 22-25 – The two sentences seem to be saying the same thing. Lines 24-25 can be reworded so the sentences do not duplicate each other.
- The Introduction is not established and there needs to be a more compelling rationale for this study. The rationale for choosing this topic, setting, and research approach needs to be further asserted. Also, it needs to be stated on how this study adds to the existing literature. Is there a gap in the literature that this study addresses? Are there similar studies that have been conducted on this topic?
- There needs to be more insight about the sociodemographic characteristics of Poland to help inform the reader and provide context for this study.
- The information on the study design is not thorough. There should be further explanation on how ‘The presented study is cross-sectional in nature’ (line 71).
- Line 73 – For participants, what were the inclusion and exclusion criteria for eligibility? What was the response rate? How was the ultimate sample of 3900 participants achieved?
- For study procedures – were participants given incentives for participation?
- What was the duration of the survey?
- Lines 120-125 – Check wording in “The collected data made it possible to: 1. calculation of the average of all respondents' answers; 2. indication of attitudes towards particular social groups based on the attitude scale adopted in the study; 3. analysis of the social group in relation to the type of question and sociometric data; 4. indication of the most frequent supposed difficulties and emotions in establishing professional relations with representatives of selected social groups”.
- Line 170-171 – Check wording in “An in-depth analysis of the respondents' statements revealed statistically significant relation-ship between the some variables of social group…”.
- In the Discussion, some of the information (lines 216-221, lines 234-238) could be included in the Introduction to better establish a rationale for this study.
- The author should consider discussing future directions for research and policy implications of the study should
- The paper should be checked for English language and style.
Overall, this is a unique study on a pertinent topic. The background and significance, procedures, methods, discussions, and conclusions of the paper need to be further developed. Tending to these items may help to clarify and improve the paper.
Author Response
Dear Reviewer,
Thank you for your suggestions. I have tried to adapt to them.
I hope the manuscript looks better now. I have highlighted in yellow all the changes made.
In the paper title, consider a colon between the two parts of the title. For instance: ‘Nurses’ Attitudes Towards Selected Social Groups: Cross-sectional survey among Polish nurses’.
I corrected it. Thank you for your suggestion.
Line 15 – Check spelling in “sur-vey”
I corrected it. Thank you for drawing attention to this oversight.
Lines 22-25 – The two sentences seem to be saying the same thing.
I corrected it. Thank you for drawing attention to this oversight.
Lines 24-25 can be reworded so the sentences do not duplicate each other.
I corrected it. Thank you for drawing attention to this oversight.
The Introduction is not established and there needs to be a more compelling rationale for this study. The rationale for choosing this topic, setting, and research approach needs to be further asserted. Also, it needs to be stated on how this study adds to the existing literature. Is there a gap in the literature that this study addresses? Are there similar studies that have been conducted on this topic?
Thank you for your suggestions. I corrected this part of manusctript.
There needs to be more insight about the sociodemographic characteristics of Poland to help inform the reader and provide context for this study.
Thank you for your suggestions. I corrected this part of manusctript
The information on the study design is not thorough. There should be further explanation on how ‘The presented study is cross-sectional in nature’ (line 71).
I changed that to a new sentence: A cross-sectional study design was used. Thank you for your opinion. Now it is clearer to the reader
Line 73 – For participants, what were the inclusion and exclusion criteria for eligibility? What was the response rate? How was the ultimate sample of 3900 participants achieved?
Thank you for your suggestions. I corrected this part of manusctript
For study procedures – were participants given incentives for participation?
I added: No incentive for participation was given.
What was the duration of the survey?
I added: The time taken to respond and return the answers was 15 minutes.
You can find in manuscript also: Data collection took place between April and July 2021.
Lines 120-125 – Check wording in “The collected data made it possible to: 1. calculation of the average of all respondents' answers; 2. indication of attitudes towards particular social groups based on the attitude scale adopted in the study; 3. analysis of the social group in relation to the type of question and sociometric data; 4. indication of the most frequent supposed difficulties and emotions in establishing professional relations with representatives of selected social groups”.
These statements have been deleted as they are not directly relevant to the analysis
Line 170-171 – Check wording in “An in-depth analysis of the respondents' statements revealed statistically significant relation-ship between the some variables of social group…”.
I corrected it. Thank you for drawing attention to this oversight.
In the Discussion, some of the information (lines 216-221, lines 234-238) could be included in the Introduction to better establish a rationale for this study.
Thank you for your suggestions. I corrected this part of manustript
Reviewer 4 Report
- What is a Roma person? it should be specified.
- You do not include in the methods any information about sociodemographic and laboral data collection.
- Did you include any information before the questionnaire indicating that it was only for nurses?
- How many items that the questionnaire has? It would be good to include it as Supplementary material.
- Include the mean age of the respondents.
- Figure 1 would be better as a table rather than a figure.
- Why do you mix the "hearing impaired" or "sexual orientation" with the ethnicity and religion in the questions? That can alter the results. I think that the questionnaire is not quite reliable for measuring what you want to measure.
- p = 0.00 is not correct. It should be p < 0.01
Author Response
Dear Reviewer,
Thank you for your suggestions. I have tried to adapt to them.
I hope the manuscript looks better now. I have highlighted in yellow all the changes made.
Round 2
Reviewer 3 Report
The authors have done well to respond to the reviewer feedback and should be commended for their work.
There are additional items for consideration:
- Lines 52-56 – There should be a citation to support this information, particularly since some of the words in this paragraph are placed in quotes.
- Line 70 – Check spelling in “In the literature, one can find described attitudes, i.n. nursing students or social work…”. There are two periods in the word ‘in’.
- Lines 72-23 – In the sentence “However, there are no studies showing a comparison of the attitudes of nurses towards different social groups…”, this seems to be a sweeping statement. Either the authors can provide a citation to support the statement or authors can temper/soften the statement along the lines of ‘there are limited studies that show a comparison…’. Either way, a citation should be included to support the statement.
- Lines 78-79 – In “It can also result in patients being unequally treated and leaving them with anxious needs”, provide a citation to support this statement.
- Lines 90-92 – In “According to statistics from the Polish Association of the Deaf, there are approximately 800,000 people with hearing impairments in Poland, including approximately 50,000”, this sentence is incomplete. Approximately 50,000 of what?
- Line 122 – In “the terrorist attacks in Europe have intensified these debates”, how is ‘terrorist attacks’ defined? Who are carrying out the ‘terrorist attacks’? One could argue that terrorist attacks are not limited to Muslims and can be carried out my members of any group, including the majority group in a society.
- Lines 88-128 – Overall, the added information is very insightful and comprehensive and helps to better support a rationale for this paper.
- Lines 131-132 – Information on the study design is still lacking. What is meant by a “diagnostic survey method” and “a self-constructed survey questionnaire”?
- Line 155 – In “To avoid multiple participation in the survey posted on social networking and discussion groups, the title of the survey was provided at the beginning of the questionnaire…”, what is meant by multiple participation? How was multiple participation avoided?
- Line 201 – Place a period at the end of this sentence.
- Lines 247-249 – In “Results The single sample chi-square test of variance showed that the respondents would not accept a homosexual as a close relative, and in the case of Muslims, the lack of acceptance concerned all the situations discussed in the study (Table. 5)”, make this into one sentence. For example, “Results in Table 5 show…”.
- Lines 284-286 – In “Contemporary European societies, not only Poland, are no longer homogeneous societies in dimension and understanding, whose internal cohesion is created and strengthened by a uniform identity, history, axiology, and culture”, the first part of the sentence conflicts with the second part. If you say no longer homogenous, this is conflicted by saying such societies’ internal cohesion was strengthen by homogeneity. Consider removing the word ‘strengthened’ to bolster this point. For example, “Contemporary European societies, not only Poland, are no longer homogeneous societies in dimension and understanding, whose internal cohesion is created by a uniform identity, history, axiology, and culture”.
- Lines 331-335 and 339-343 – This is insightful information.
- Lines 343-345 – In “They showed those who have a stronger attitude towards Roma, and the respective stereotypical, emotional and behavioral assessments are much more causally related”, this sentence is unclear. What are respective stereotypical, emotional, and behavioral assessments? What is meant by causally related?
- Lines 359-365 – This is insightful information.
- Line 384 – In “ cheaper thanks to the Internet”, check for casual and informal working.
Overall, the revised manuscript is more detailed and compelling. There are still items to be tended to help make the paper clearer. Incorporating the additional feedback may help to improve the paper and make it suitable for publication.
Author Response
Dear Reviewer,
Thank you for your comments and suggestions. I have tried to improve the manuscript according to your feedback. I hope that it is now readable and does not raise your concerns.
I have highlighted in yellow all the corrections made.
Please find find below answers to your suggestions:
Lines 52-56 – There should be a citation to support this information, particularly since some of the words in this paragraph are placed in quotes.
Thank you for your suggestion. I have added source of words
Line 70 – Check spelling in “In the literature, one can find described attitudes, i.n. nursing students or social work…”. There are two periods in the word ‘in’.
Thank you for your suggestion. I have corrected it.
Lines 72-23 – In the sentence “However, there are no studies showing a comparison of the attitudes of nurses towards different social groups…”, this seems to be a sweeping statement. Either the authors can provide a citation to support the statement or authors can temper/soften the statement along the lines of ‘there are limited studies that show a comparison…’. Either way, a citation should be included to support the statement.
Thank you for your suggestion. I have corrected it.
Lines 78-79 – In “It can also result in patients being unequally treated and leaving them with anxious needs”, provide a citation to support this statement.
Thank you for your suggestion. I have added citation to suport this statement.
Lines 90-92 – In “According to statistics from the Polish Association of the Deaf, there are approximately 800,000 people with hearing impairments in Poland, including approximately 50,000”, this sentence is incomplete. Approximately 50,000 of what?
Thank you for your suggestion. I have removed the full stop after the number and the sentence sounds correctly.
Line 122 – In “the terrorist attacks in Europe have intensified these debates”, how is ‘terrorist attacks’ defined? Who are carrying out the ‘terrorist attacks’? One could argue that terrorist attacks are not limited to Muslims and can be carried out my members of any group, including the majority group in a society.
Thank your for your suggestion. I have added some information and I hope more clearer now
Lines 88-128 – Overall, the added information is very insightful and comprehensive and helps to better support a rationale for this paper.
Thank you for your opinion.
Lines 131-132 – Information on the study design is still lacking. What is meant by a “diagnostic survey method” and “a self-constructed survey questionnaire”?
I have removed this section because section 2.4 describes the research tool constructed for the study. I hope more clearer now.
Line 155 – In “To avoid multiple participation in the survey posted on social networking and discussion groups, the title of the survey was provided at the beginning of the questionnaire…”, what is meant by multiple participation? How was multiple participation avoided?
Thank you for your suggestion. In tools such as the one presented, repeat participation cannot be ruled out, one can only minimise the risk from the outset by informing the potential respondent about the survey. The text describes that: To minimize the possibility of the same person participating in the survey again, the survey started with a welcome screen with the title, the purpose of the survey, instructions and a link to the survey. In this way, the potential respondent knew immediately what the topic of the survey was and could leave it before starting to answer the questions. This difficulty was also mentioned in the limitations of the study.
Line 201 – Place a period at the end of this sentence.
Thank you for your suggestion. I have put it.
Lines 247-249 – In “Results The single sample chi-square test of variance showed that the respondents would not accept a homosexual as a close relative, and in the case of Muslims, the lack of acceptance concerned all the situations discussed in the study (Table. 5)”, make this into one sentence. For example, “Results in Table 5 show…”.
Thank you for your suggestion. I have corrected this sentence.
Lines 284-286 – In “Contemporary European societies, not only Poland, are no longer homogeneous societies in dimension and understanding, whose internal cohesion is created and strengthened by a uniform identity, history, axiology, and culture”, the first part of the sentence conflicts with the second part. If you say no longer homogenous, this is conflicted by saying such societies’ internal cohesion was strengthen by homogeneity. Consider removing the word ‘strengthened’ to bolster this point. For example, “Contemporary European societies, not only Poland, are no longer homogeneous societies in dimension and understanding, whose internal cohesion is created by a uniform identity, history, axiology, and culture”.
Thank you for your suggestion. I have corrected this sentence.
Lines 331-335 and 339-343 – This is insightful information.
Thank you
Lines 343-345 – In “They showed those who have a stronger attitude towards Roma, and the respective stereotypical, emotional and behavioral assessments are much more causally related”, this sentence is unclear. What are respective stereotypical, emotional, and behavioral assessments? What is meant by causally related?
Thank you for your suggestion. I have corrected this sentence and I hope more clearer now.
Lines 359-365 – This is insightful information.
Thank you
Line 384 – In “ cheaper thanks to the Internet”, check for casual and informal working.
Thank you for your suggestion. I have corrected it and I hope more clearer now.